# Vibration Suppression of Wind/Traffic/Bridge Coupled System Using Multiple Pounding Tuned Mass Dampers (MPTMD)

**DOI:** 10.3390/s19051133

**Published:** 2019-03-06

**Authors:** Xinfeng Yin, Gangbing Song, Yang Liu

**Affiliations:** 1School of Civil Engineering, Changsha University of Science & Technology, Changsha 410114, China; liuyangbridge@163.com; 2Department of Mechanical Engineering, University of Houston, Houston, TX 77204, USA; gsong@central.uh.edu

**Keywords:** wind/traffic/bridge coupled system, vibration suppression, energy dissipation, pounding tuned mass damper (PTMD), multiple pounding tuned mass dampers (MPTMDs)

## Abstract

Dynamic responses of highway bridges induced by wind and stochastic traffic loads usually exceed anticipated values, and tuned mass dampers (TMDs) have been extensively applied to suppress dynamic responses of bridge structures. In this study, a new type of TMD system named pounding tuned mass damper (PTMD) was designed with a combination of a tuned mass and a viscoelastic layer covered delimiter for impact energy dissipation. Comprehensive numerical simulations of the wind/traffic/bridge coupled system with multiple PTMDs (MPTMDs) were performed. The coupled equations were established by combining the equations of motion of both the bridge and vehicles in traffic. For the purpose of comparing the suppressing effectiveness, the parameter study of the different numbers and locations, mass ratio, and pounding stiffness of MPTMDs were studied. The simulations showed that the number of MPTMDs and mass ratio are both significant in suppressing the wind/traffic/bridge coupled vibration; however, the pounding stiffness is not sensitive in suppressing the bridge vibration.

## 1. Introduction

The bridge vibrations under wind and stochastic traffic loads have been extensively studied, and great successes have been achieved during the recent decades. The dynamic performance of bridges is affected by many factors, such as the wind speed, vehicle type, and road surface condition [1,2,3,4,5]. The results of all above research have shown that the dynamic responses of bridges induced by wind and stochastic traffic loads usually exceed that anticipated on highway bridges. 

To control the dynamic response of the primary structure, a method of suppressing the vibration of structures is to use an energy dissipative system. One such system is the tuned mass damper (TMD) system, which is used as a secondary vibration system and is connected to the primary structure at suitable positions. Since the TMD concept was first investigated by Frahm [6], much research has been carried out to examine its effectiveness for various dynamic load applications. Much of the previous research efforts were focused on optimizing the TMD parameters and suppressing the bridge vibration under wind or seismic loads [7,8,9,10,11,12,13,14,15]. However, we found few studies on applying TMDs to suppress bridge vibrations induced by combined wind and traffic loads [16,17]. Recently, a new damping system, pounding tuned mass dampers (PTMD), was proposed and obtained from the references [18,19,20]. In the PTMD design, the damper has a moving mass block (which is similar to that of a traditional TMD) and an additional delimiter covered with viscoelastic material to restrict the stroke of the mass block and dissipate energy via impacts or pounding [19,20,21]. Compared with a traditional TMD system, the PTMD can simultaneously suppress vibrations in two directions with easy installation and maintenance [18,19]. In addition, the PTMD has larger energy dissipating capacity through impact and is more robust to the system uncertainty compared with a regular TMD [20,21,22]. However, to the best of the authors’ knowledge, very few studies were found on applying PTMDs to suppress the vehicle-induced bridge vibrations. 

In this study, a multiple PTMD (MPTMD) system was designed for suppression of wind and traffic induced vibration of a highway bridge. A comprehensive numerical simulation of the traffic/bridge/MPTMD coupled system was performed. The wind/traffic/bridge/MPTMD coupled equations were established by combining the equations of motion of both the bridge and vehicles in traffic, where the displacement and interaction force relationship between the tire and bridge surface roughness was used. For the purpose of comparing the suppressing effectiveness, the parameter study of the different numbers and locations, mass ratio, and pounding stiffness of MPTMD were studied. 

## 2. Methodology of the Traffic-Bridge Coupled System with MPTMD

### 2.1. Equivalent Vehicle Model Approach in Traffic

As discussed by Liu et al. [23], the traffic-bridge coupled system can consider various types and numbers of vehicles at any location on the bridge. In order to simplify the vehicular models in traffic, all the vehicles are classified as three types: (1) heavy multi-axle trucks; (2) light trucks and buses; and (3) sedan cars. Only heavy trucks are modeled with 18-DOF(Degree of Freedom) full scale vehicle model, and the other light trucks and sedan cars are used with the 3-DOF vehicle model to be computationally efficient. The 18-DOF and 3-DOF vehicle models are shown in Figure 1 and Figure 2, respectively. The detailed vehicle parameters can be obtained from Yin et al. [24,25]. 

### 2.2. Equation of Motion of the Vehicle

The equation of motion for a vehicle can be expressed as follows: (1)$$\left[M_{v}\right]\left\{{\ddot{Y}}_{v}\right\}+\left[C_{v}\right]\left\{{\dot{Y}}_{v}\right\}+\left[K_{v}\right]\left\{Y_{v}\right\}=\left\{F_{G}\right\}+\left\{F_{v\text{-}p}\right\}+\left\{F_{vw}\right\}$$where $\left[M_{v}\right]$, $\left[C_{v}\right]$, and $\left[K_{v}\right]$ are the mass, damping, and stiffness matrices of the vehicle, respectively; $\left\{Y_{v}\right\}$ is the vector including the displacements of the vehicle; $\left\{{\ddot{Y}}_{v}\right\}$, $\left\{{\dot{Y}}_{v}\right\}$ are respectively the vector including the acceleration and velocity of the vehicle; $\left\{F_{G}\right\}$ is the gravity force vector of the vehicle; $\left\{F_{v\text{-}p}\right\}$ is the vector of the wheel-pavement contact forces acting on the vehicle; and $\left\{F_{\mathrm{vw}}\right\}$ is the vector of the wind forces acting on the vehicle [5]. 

### 2.3. Equation of Motion of the Bridge

The structural nonlinearity is typically considered in the static analysis before a dynamic analysis is conducted. Equations of motions in the three directions including vertical, lateral, and torsion of the bridge with the mode superposition technique can be obtained in a matrix form as [2]: (2)$$\left[M_{b}\right]\left\{{\ddot{Y}}_{b}\right\}+\left[C_{b}\right]\left\{{\dot{Y}}_{b}\right\}+\left[K_{b}\right]\left\{Y_{b}\right\}=\left\{F_{b\text{-}v}{+F}_{b\text{-}p}{+F}_{\mathrm{bw}}\right\}$$where $\left[M_{b}\right]$, $\left[C_{b}\right]$, and $\left[K_{b}\right]$ are the mass, damping, and stiffness matrices of the bridge, respectively; $\left\{Y_{b}\right\}$ is the displacement vector for all DOFs of the bridge; $\left\{{\dot{Y}}_{b}\right\}$ and $\left\{{\ddot{Y}}_{b}\right\}$ are the first and second derivative of $\left\{Y_{b}\right\}$ with respect to time, respectively; $\left\{F_{b\text{-}v}\right\}$ is a vector containing all external forces acting on the bridge; $\left\{F_{b\text{-}p}\right\}$ is the interacting force between the PTMD and the bridge, and $\left\{F_{\mathrm{bw}}\right\}$ is the vector of the wind forces acting on the bridge. 

### 2.4. Basics about PTMD and Introduction to MPTMD

In the previous study of Song et al. [20], the comparison of the schematic of a TMD system, an impact damper, and the PTMD system was given. In the PTMD design in this paper, the damper has a moving mass block, which is similar to that of a regular TMD, and a delimiter is covered with viscoelastic material to restrict the stroke of the mass block and dissipate energy via impacts or pounding [19,20]. Figure 3 and Figure 4 illustrate MPTMD designed for this purpose. It can be seen that the PTMD consists of two parts—the TMD and the delimiter covered with viscoelastic material. For the multiple PTMD design, the MPTMD with *n* number of parallelly-placed PTMDs can be installed on a bridge structure at section *x* = *x_s_*, as shown in Figure 4. To obtain the effectiveness of suppressing vibration and the parameters of the pounding force, the small-scale PTMD equipment was designed and used to control vibration a pipeline structure. More details on the experimental setup can be obtained from the Song et al. [20] and Zhang et al. [18].

### 2.5. Equations of Motion of the MPTMD

The equations of motion that represent the interaction between each PTMD and the bridge are: (3)$$m_{p}{\ddot{y}}_{pv}(t)+c_{p}{}_{v}{\dot{y}}_{pv}(t)+k_{p}{}_{v}y_{pv}(t)=-f_{p\text{-}b}{}^{v}(t)-Hf_{p\text{-}b}{}^{vp}(t)$$
(4)$$m_{p}{\ddot{y}}_{pl}(t)+c_{pl}{\dot{y}}_{pl}(t)+k_{pl}y_{pl}(t)=-f_{p\text{-}b}{}^{l}(t)-Hf_{p\text{-}b}{}^{lp}(t)$$
(5)$$\left[M_{p}\right]\left\{{\ddot{U}}_{p}\right\}+\left[C_{p}\right]\left\{{\dot{U}}_{p}\right\}+\left[K_{p}\right]\left\{U_{p}\right\}=\left\{F_{p\text{-}b}\right\}+H\Gamma \left\{F_{p\text{-}b}{}^{p}\right\}$$
where $m_{p}$ is the mass of the PTMD; $c_{pv}$ and $c_{pl}$ are the damping of the PTMD in the vertical and lateral direction; $k_{p}{}_{v}$ and $k_{pl}$ are the stiffness of the PTMD; $f_{p\text{-}b}{}^{l}(t)$ is the force produced by the relative motion between the bridge and the PTMD in the lateral direction; $f_{p\text{-}b}{}^{v}(t)$ denotes that force in the vertical direction. $f_{p\text{-}b}{}^{vp}(t)$ and $f_{p\text{-}b}{}^{lp}(t)$ are the pounding forces in the vertical direction and horizontal direction, which can be calculated using Equations (3)–(5), and $\left[M_{p}\right]$, $\left[C_{p}\right]$, and $\left[K_{p}\right]$ are the mass, damping, and stiffness matrices of the PTMD, respectively; $\left\{U_{p}\right\}$, $\left\{{\ddot{U}}_{p}\right\}$, $\left\{{\dot{U}}_{p}\right\}$ are the displacement, acceleration, and velocity vector for all DOFs of the PTMD, and the variable *H* describes the direction of the pounding force; Γ denotes the location of the pounding forces. 

### 2.6. Model of the Pounding Force

A numerical model is required to accurately analyze the dynamic response of the bridge structure controlled by a PTMD. Several models were proposed to study the effect of the PTMD on the major structure in dynamic analysis during the past decade. Among them, the nonlinear model based on the Hertz contact element in conjunction with a damper is the most appropriate [22] and is thus adopted in this paper. 

The force produced by the relative motion between the bridge and the PTMD is expressed as:
(6)$$f_{p\text{-}b}{}^{v}(t)=k_{pv}[y^{v}{}_{b}-y^{v}{}_{p}];\;f_{p\text{-}b}{}^{l}(t)=k_{pl}[y^{l}{}_{b}-y^{l}{}_{p}]$$

The pounding force is expressed as:
(7)$$f_{p\text{-}b}{}^{vp}(t)=\{\begin{array}{ll} \beta {\left(x_{1}-x_{2}-g_{p}\right)}^{\frac{3}{2}}+c({\dot{x}}_{1}-{\dot{x}}_{2}), & x_{1}-x_{2}-g_{p}>0,{\dot{x}}_{1}-{\dot{x}}_{2}>0;\\ \beta {\left(x_{1}-x_{2}-g_{p}\right)}^{\frac{3}{2}}, & x_{1}-x_{2}-g_{p}>0,{\dot{x}}_{1}-{\dot{x}}_{2}<0;\\ 0, & x_{1}-x_{2}-g_{p}<0 \end{array}$$
where $x_{1}$ and $x_{2}$ are the displacements of the pounding motion limiting collar and the mass block, and $g_{p}$ is the gap between them. $x_{1}\text{-}x_{2}\text{-}g_{p}$ denotes the relative pounding deformation, and ${\dot{x}}_{1}\text{-}{\dot{x}}_{2}$ is the relative velocity. β is the pounding stiffness coefficient that mainly depends on material properties and the geometry of colliding bodies. Since the viscoelastic material is highly nonlinear, the impact damping *c* is not a constant. *c* depends on the pounding stiffness and the deformation of the viscoelastic layer. At any instant of time, *c* can be obtained from Equation (8) [26]: (8)$$c=2\xi \sqrt{\beta \sqrt{x_{1}-x_{2}-g_{p}}\frac{m_{1}m_{2}}{m_{1}+m_{2}}}$$
(9)$$\xi =\frac{9\sqrt{5}}{2}\frac{1-e^{2}}{e(e(9\pi -16)+16)}$$
where $m_{1}$ and $m_{2}$ are the mass of the two colliding bodies, and ξ is the impact damping ratio correlated with the coefficient of restitution *e*, which is defined as the relation between the post impact (final) relative velocity. When *e* = 0 stands for a perfectly plastic impact, *e* = 1 stands for a fully elastic impact. 

After assessing the value of ξ, the pounding stiffness β can be determined numerically through interactive simulation, which fits the experimentally obtained pounding force time histories. 

### 2.7. Assembly of the Traffic-Bridge Coupled System with MPTMD

Using the displacement relationship and the interaction force relationship at the contact points between the bridge and vehicles in traffic, the traffic-bridge coupled system with MPTMD can be established by combining the equations of motion of both the bridge and vehicles, as shown below:
(10)$$\begin{array}{l} \left[\begin{array}{ccc} M_{b} & & \\ & M_{p} & \\ & & M^{N}{}_{v} \end{array}\right]\left\{\begin{array}{c} {\overset{¨}{Y}}_{b}\\ {\overset{¨}{Y}}_{p}\\ {\overset{¨}{Y}}_{v} \end{array}\right\}+\left[\begin{array}{ccc} C_{b}+C_{bb}+C_{p\text{-}b} & C_{b\text{-}p} & C_{b\text{-}v}\\ C_{p\text{-}b} & C_{p}+C_{p\text{-}p} & 0\\ C_{v\text{-}b} & 0 & C_{v}{}^{N}+C_{v\text{-}v}{}^{N} \end{array}\right]\left\{\begin{array}{c} {\dot{Y}}_{b}\\ {\dot{Y}}_{p}\\ {\dot{Y}}_{v} \end{array}\right\}+\\ \left[\begin{array}{ccc} K_{b}+K_{bb}+K_{p\text{-}b} & K_{b\text{-}p} & K_{b\text{-}v}\\ K_{p\text{-}b} & K_{p}+K_{p\text{-}p} & 0\\ K_{v\text{-}b} & 0 & K_{v}{}^{N}+K_{v\text{-}v}{}^{N} \end{array}\right]\left\{\begin{array}{c} {\overset{}{Y}}_{b}\\ {\overset{}{Y}}_{p}\\ {\overset{}{Y}}_{v} \end{array}\right\}==\left\{\begin{array}{c} F^{N}{}_{bw}+F^{N}{}_{b\text{-}v}\\ F^{N}{}_{p\text{-}b}+H\Gamma F^{p}{}_{p\text{-}b}\\ F^{N}{}_{v\text{-}b}+F_{G}{}^{N}+F^{N}{}_{vw} \end{array}\right\} \end{array}$$
where N is the number of vehicles traveling on the bridge; $M_{v}{}^{N}$, $C_{v}{}^{N}$, and $K_{v}{}^{N}$ are mass, damping, and stiffness matrices for the vehicle, respectively; $C^{N}{}_{b\text{-}vb}$ and $K^{N}{}_{b\text{-}vb}$ are damping and stiffness contribution to the bridge structure due to the coupling effects between the N vehicles in the vehicle and the bridge system, respectively; $C^{N}{}_{b\text{-}v}$ and $K^{N}{}_{b\text{-}v}$ are the coupled stiffness and damping matrices contributing to bridge vibration from the N vehicles in traffic, respectively; $C^{N}{}_{v\text{-}b}$ and $K^{N}{}_{v\text{-}b}$ are the coupled damping and stiffness matrices contributing to the vibration of the N vehicles, respectively; $C_{v\text{-}v}{}^{N}$ and $K_{v\text{-}v}{}^{N}$ are the coupled damping and stiffness matrices of induced by other vehicles, respectively. Equation (10) can be solved by the improved New-mark method in the time domain. 

### 2.8. Method of Evaluating Ride Comfort

To evaluate the ride comfort, the ISO2631-1 [27] specifies the root-mean-square (RMS) magnitudes of the vibration acceleration as the standard for ride comfort, as shown in Table 1. 

For vibrations in more than one direction, the weighted RMS acceleration $a_{w}$ determined from the vibrations in the orthogonal coordinates is calculated as: (11)$$a_{w}={\left(k_{ax}{}^{2}a_{wx}{}^{2}+k_{ay}{}^{2}a_{wy}{}^{2}+k_{az}{}^{2}a_{wz}{}^{2}\right)}^{\frac{1}{2}}$$where $a_{wx}$, $a_{wy}$, and $a_{wz}$ are the weighted RMS accelerations with respect to the orthogonal axes *x*, *y*, and *z*, respectively; $k_{ax}$, $k_{ay}$ and $k_{az}$ are multiplying factors with the orthogonal axes *x*, *y*, and *z*, respectively, and: (12)$${a_{wj}|}_{j=x,y,z}={\left[\frac{1}{T}\int_{t=0}^{t=T}{a^{2}{}_{wj}(t)|}_{j=x,y,z}dt\right]}^{\frac{1}{2}}$$where $a_{wj}\left(t\right)|\;$is the acceleration as a function of time (m/s^2^) in the *x*, *y*, and *z* axes directions; and T is the duration of the measurement (s). 

## 3. Numerical Studies

### 3.1. Description of an Existing Bridge

A typical high-pier bridge is shown in Figure 5. The bridge is a five-span, two-lane straight continuous beam bridge with 812 m in length and 12.5 m in width. The highest pier measures 178 m in height. Figure 6 shows the overview of the bridge and the cross-section of the bridge. The Finite Element (FE) model of the high-pier Bridge is shown in Figure 7. 

### 3.2. The Parameters of Vehicles in Traffic Flow

In the present study, to simplify the vehicular model, only heavy trucks are modeled with three-dimensional (3-D) vehicle models, while light trucks and sedan cars are modeled using quarter vehicle models to save computational efforts. The 3-D vehicle model and the quarter vehicle model are shown in Figure 1 and Figure 2, respectively, and the parameters of the vehicle models are summarized in Table 2 and Table 3. The mechanical and geometric properties of the test truck can be obtained from Yin et al. [24] and are listed in Table 2. The parameters of the quarter vehicle model can be obtained from reference [2] and are also shown in Table 3. 

For the purpose of traffic simulation, three different vehicle occupancy coefficients ρ are considered using Equations (1)–(3): smooth traffic (ρ=0.07), median traffic (ρ=0.15), and busy traffic flow (ρ=0.3). It is reasonably found from Table 4 that the mean speed of the traffic flow decreases while the standard deviation of the vehicle speeds increases with the increase of the vehicle occupancy. 

### 3.3. The Parameters of MPTMDs Installed on the Bridge

According to the preliminary analyses, it is desirable to tune the MPTMD to the bridge’s dominant mode in one direction, and the MPTMD are positioned at the mid-span of the each span (Figure 8) where the dynamic response is the maximum. In the case of MPTMDs, all the PTMD systems can be concentrated at the mid-span or can be distributed along the length of the bridge. Figure 8 shows the simplified model of the bridge with MPTMD attached at equal intervals under the bridge. Figure 9b shows the N locations of parallelly-placed single PTMDs installed with equal or unequal intervals on the bridge at sections I-I (II-II; III-III), and Figure 8c shows the N numbers of single PTMD located at cross-section in a lateral direction. 

In the studies [9,16,17], the damping ratios of all the bridges are assumed to be 0.02. The mass ratio of both MPTMDs is selected as 1 % in this study. The coefficient *e* of restitution of the each PTMD is set as 0.5, and the pounding stiffness β of the PTMD is $25,000{\mathrm{Nm}}^{-3/2}$. The MPTMD system can own a wider frequency range, and each PTMD contains respective adjustable natural frequency, as shown in Table 5. 

To compare the performance of the MPTMD with different parameters, the vibration reduction ratio is defined as: (13)$$\eta_{{}_{Ctrl}}=\frac{Y_{O}-Y_{Ctrl}}{Y_{O}}\times 100\%$$where $Y_{O}$ and $Y_{Ctrl}$ are the maximum displacement of the coupled system without and with PTMD.

## 4. Numerical Simulations

### 4.1. Comparison of the Bridge Responses with Different Traffic Flow Occupancies

As the preliminary analyses of MPTMDs on the wind/traffic/bridge/MPTMD coupled vibration, the bridge responses with different traffic flow occupancies are studied firstly in this section. The time histories of the vertical and lateral responses at the mid-span of the bridge under three situations with a single vehicle and two types of traffic flow occupancies are presented in Figure 9. It was found that both vertical and lateral displacements at the mid-span increased generally as the vehicle occupancy increased. The vehicle occupancies played a significant role in the response of bridge displacements. For example, Figure 9a shows that the maximal vertical displacements of the bridge increased from 21.92 mm to 35.29 mm when the vehicle occupancy changed from the smooth traffic to the median traffic. 

### 4.2. Comparison of the Bridge Responses under Median Traffic Flow with Different Wind Speeds

The time histories of the responses at the mid-span of the bridge under two typical wind speeds (weak wind speed *U* = 2.7 m/s and moderate wind speed *U* = 17.6 m/s) with the median traffic flow occupancy ρ=0.15 are presented in Figure 10. It was found that the displacements at the mid-span increased generally with the increase of wind speeds, as expected. The wind speed played a significant role in the bridge displacements. For example, when the wind speed increased from 2.7 m/s to 17.6 m/s, the maximal vertical displacements increased from 56.93 mm to 73.59 mm, and the maximal lateral displacements increased from 41.76 mm to 51.60 mm. 

### 4.3. Study of MPTMDs on the Wind/Traffic/Bridge/MPTMD Coupled Vibration

As discussed above, the vertical and lateral displacements induced by traffic are both significant and thus cannot be neglected. Therefore, the vertical and lateral dynamic displacements should be suppressed by MPTMD systems, and the MPTMD systems are tuned respectively to vertical frequency or lateral frequency of the bridge. 

(1) Effect of the numbers and locations of MPTMD system 

The time histories of the bridge deflections without suppressing the vibration system with and without MPTMD are plotted in Figure 11 and shown in Table 6. It is evident from these figures that the PTMD were very effective in suppressing bridge vibration. The values of responses marked without suppressed systems in the figure were more than those marked by PTMD and MPTMD. Taking an example of vertical displacements of the bridge, the maximal bridge displacement was 73.59 mm for the situation without a vibration suppression system, while the maximal displacements of the bridge were 63.08 mm (14.28%) and 52.02 mm (29.31%) for situations with single PTMD and three PTMDs, respectively. Therefore, the PTMD and MPTMDs were both effective methods in suppressing the bridge vibration in the wind/traffic/bridge coupled system. Comparing the displacement values with the different numbers of MPTMDs in Table 6, the MPTMD system was more effective in reducing the bridge-forced vibration than it was when only a single PTMD system was installed under the bridge. Using the nine PTMD system, the vibration reduction ratio of the vertical displacements could reach to 36.78%. 

The accelerations of vehicle body and ride comfort are given in Figure 12 and Table 7, respectively. It can be seen that the MPTMDs were very effective in suppressing vehicular acceleration. The values of vehicular acceleration marked without suppressed system in the figure were much more than those marked by MPTMDs. In addition, comparing the acceleration values shown in the Table 7, it can be seen that MPTMD systems were an effective method in the situation of reducing the ride comfort, and the effectiveness of the MPTMD system was much better than a single PTMD system. 

(2) Effect of mass ratio of MPTMD system 

The mass ratio is one of the key parameters in changing the reduction performance. For economic reasons, the mass ratio is usually from 0.5% to 2%. As illustrated in Figure 13 and Table 8 and Table 9, the vibration reduction ratio of vertical displacements increased significantly, from 21.09% to 45.47%, with the mass ratio increasing from 0.5% to 2%, and the ride comfort varied from “uncomfortable” to “not uncomfortable”. 

(3) Effect of pounding stiffness of MPTMD system 

The pounding stiffness β is also a key parameter in modeling the pounding force. It is determined by material properties and geometry of colliding bodies. It varies from 10,000 to 30,000 according to the durability test [18]. In this section, the pounding stiffness was changed to 5000 and 35,000. Other parameters such as the PTMD’s mass ratio and number of MPTMDs remained the same. Figure 14 and Figure 15 and Table 10 and Table 11 show the effect of pounding stiffness on the bridge displacements and vehicle accelerations. It is seen that the effectiveness of MPTMD control performance was not very sensitive to the change of pounding stiffness. As the pounding stiffness increased from 5000 to 35,000, the reduction ratio was only decreased from 31.21% to 27.94%. 

## 5. Conclusions

In the present study, a new PTMD system was designed with the integration of a tuned mass and a viscoelastic-material covered delimiter for energy dissipation during impacts. A comprehensive numerical simulation of the wind/traffic/bridge/MPTMD coupled system was performed with consideration to the effects of the traffic flow model, the MPTMD systems, and wind forces. The wind/traffic/bridge/MPTMD coupled equations were established by combining the equations of motion of both the bridge and vehicles in traffic, and the displacement and interaction force relationship between the tire and bridge surface roughness was used. For the purpose of comparing the suppressing effectiveness, the parametric study of the different numbers and locations, mass ratio, and pounding stiffness of MPTMD were conducted. The numerical simulations demonstrated that: 

The single PTMD and multiple PTMD were both very effective in suppressing vehicular acceleration and bridge displacements induced by the traffic flows in the wind environment. The effectiveness of the multiple PTMD system was much better than that of the single PTMD system. The number of MPTMDs was significant in suppressing the bridge vibration in the wind/traffic/bridge coupled system. Compared with the displacement values under the different numbers of the MPTMD system, the vibration reduction ratio of the vertical displacements could reach 36.78%. The effects of vibration reduction ratio on responses of wind/traffic/bridge coupled system were the mass ratio increasing from 0.5% to 2%, the bridge displacement increasing significantly from 21.09% to 45.47%, and the ride comfort varying from “uncomfortable” to “not uncomfortable”. The effectiveness of MPTMD control performance was not very sensitive to the change of pounding stiffness. As the pounding stiffness increased from 5000 to 35,000, the reduction ratio was only decreased from 31.21% to 27.94%. 

## Figures and Tables

**Figure 1 sensors-19-01133-f001:**
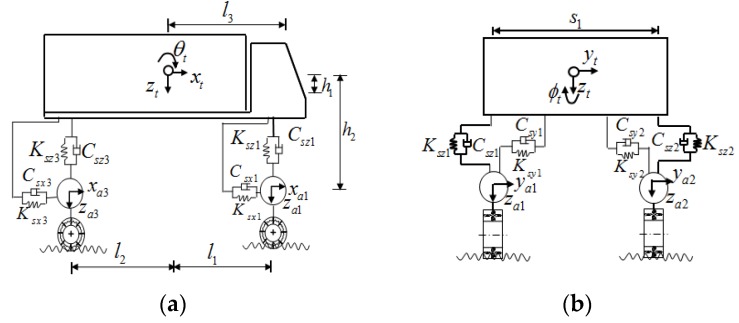
An 18-DOF full-scale vehicle model. (**a**) Elevation view. (**b**) Cross-section view.

**Figure 2 sensors-19-01133-f002:**
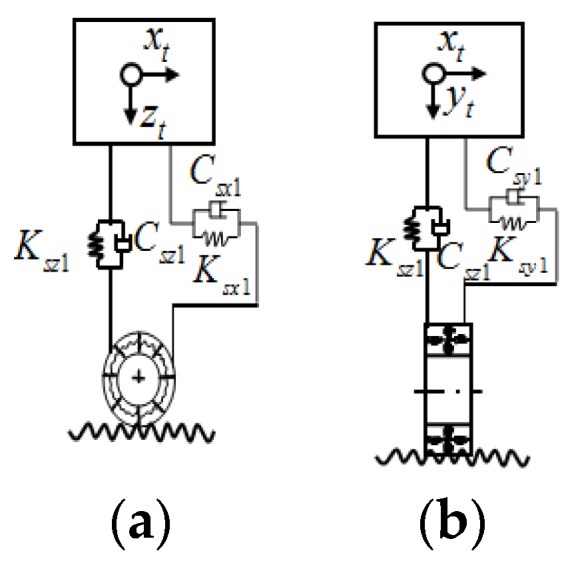
A 3-DOFs vehicle model with three-dimensional (3-D) vibrations. (**a**) Elevation view. (**b**) Cross-section view.

**Figure 3 sensors-19-01133-f003:**
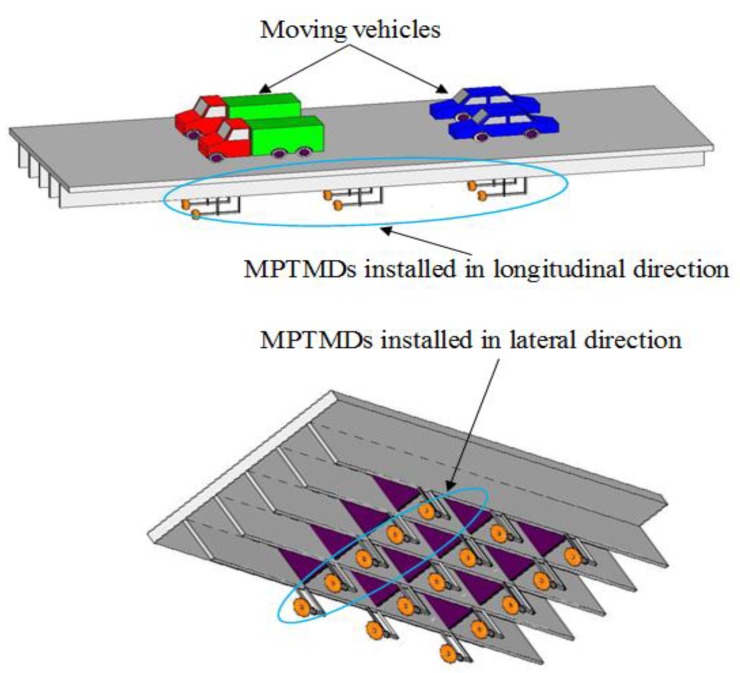
Traffic-bridge coupled system with multiple pounding tuned mass dampers (MPTMDs).

**Figure 4 sensors-19-01133-f004:**
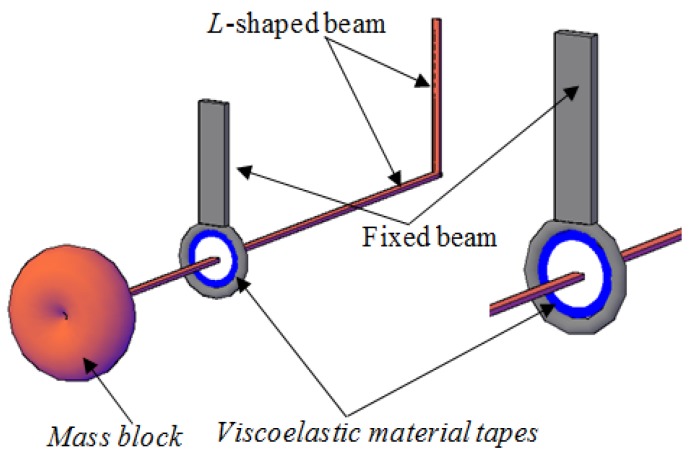
Components of a pounding tuned mass damper (PTMD) system.

**Figure 5 sensors-19-01133-f005:**
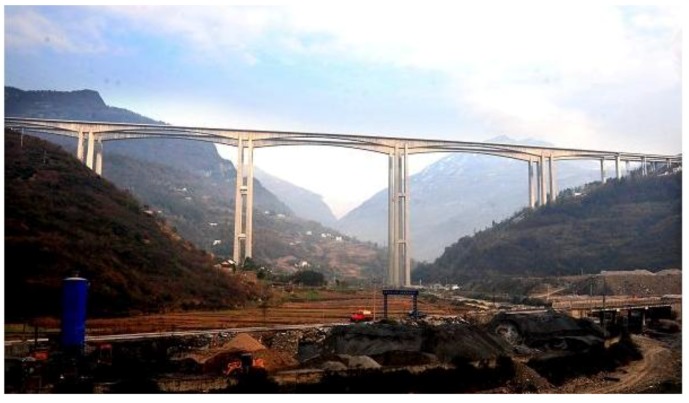
A typical high-pier bridge.

**Figure 6 sensors-19-01133-f006:**
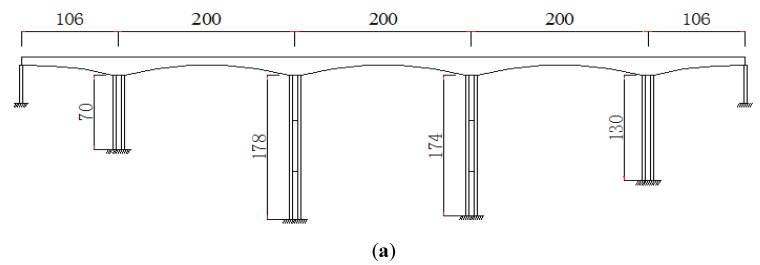
Elevation of the bridge and the cross-section of the beam. (**a**) Elevation of the high-pier bridge (m). (**b**) Cross-section of the beam (m).

**Figure 7 sensors-19-01133-f007:**
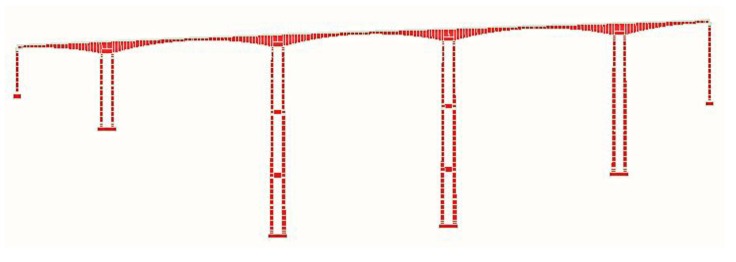
FE model of the Longtan Bridge.

**Figure 8 sensors-19-01133-f008:**
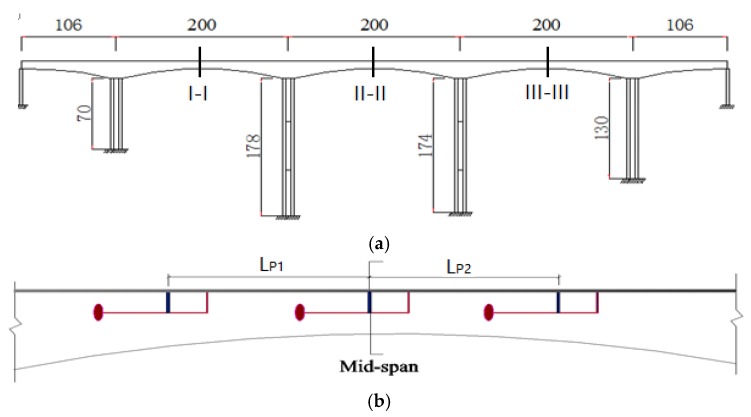
Locations of the MPTMD at the bridge. (**a**) Locations of the MPTMD at each span (m). (**b**) Longitudinal locations of the MPTMD at each mid-span (I-I; II-II; III-III) (m). (**c**) Lateral locations of the MPTMD at each mid-span (I-I; II-II; III-III) (m).

**Figure 9 sensors-19-01133-f009:**
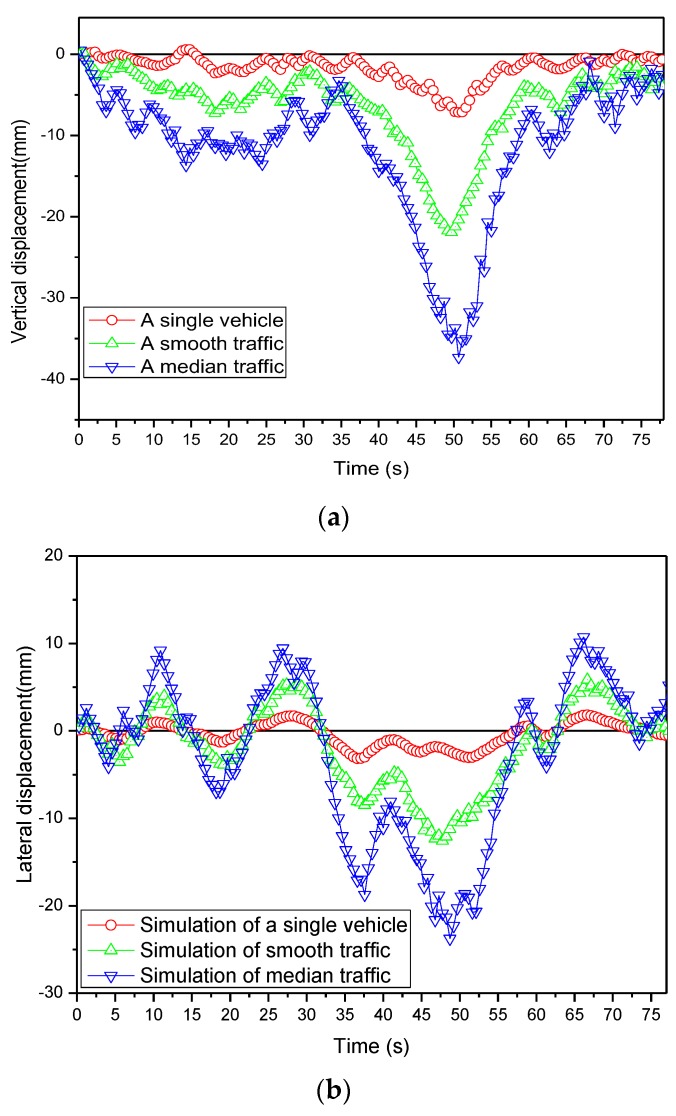
Comparison of simulated solutions of the mid-span displacement. (**a**) Vertical displacement. (**b**) Lateral displacement.

**Figure 10 sensors-19-01133-f010:**
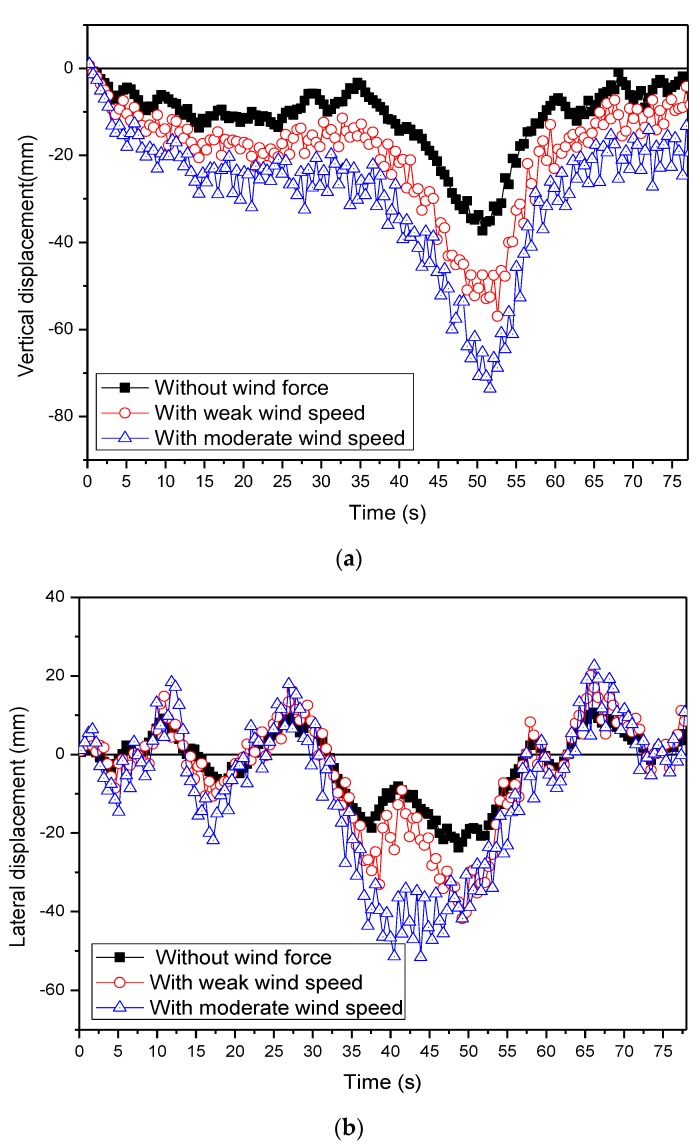
Comparison of simulated solutions of the mid-span displacement. (**a**) Vertical displacement (**b**) Lateral displacement.

**Figure 11 sensors-19-01133-f011:**
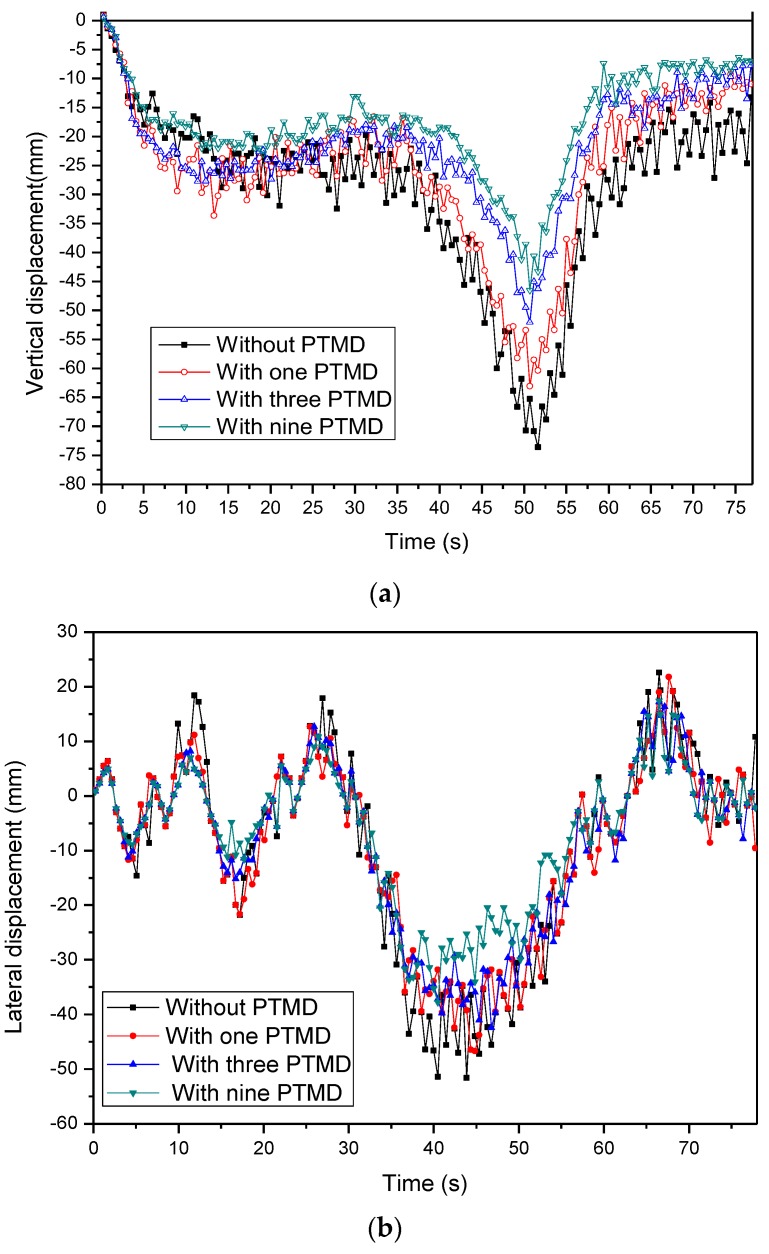
Comparison of the mid-span displacements with/without MPTMDs. (**a**) Vertical displacement. (**b**) Lateral displacement.

**Figure 12 sensors-19-01133-f012:**
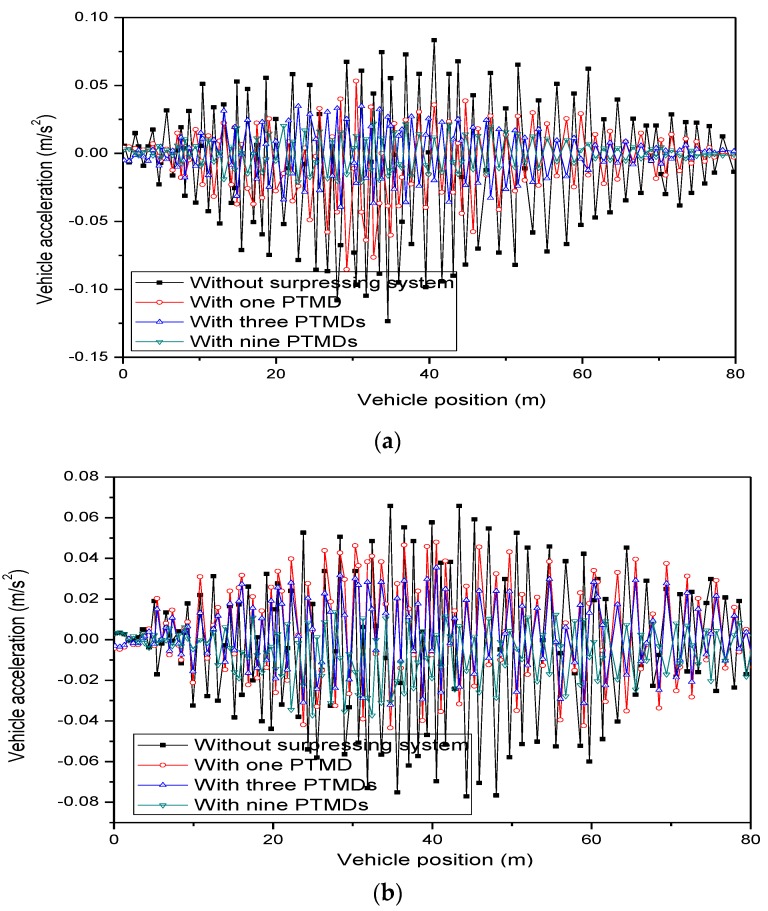
Comparison of the vehicle accelerations with/without MPTMDs. (**a**) Vertical accelerations. (**b**) Lateral accelerations.

**Figure 13 sensors-19-01133-f013:**
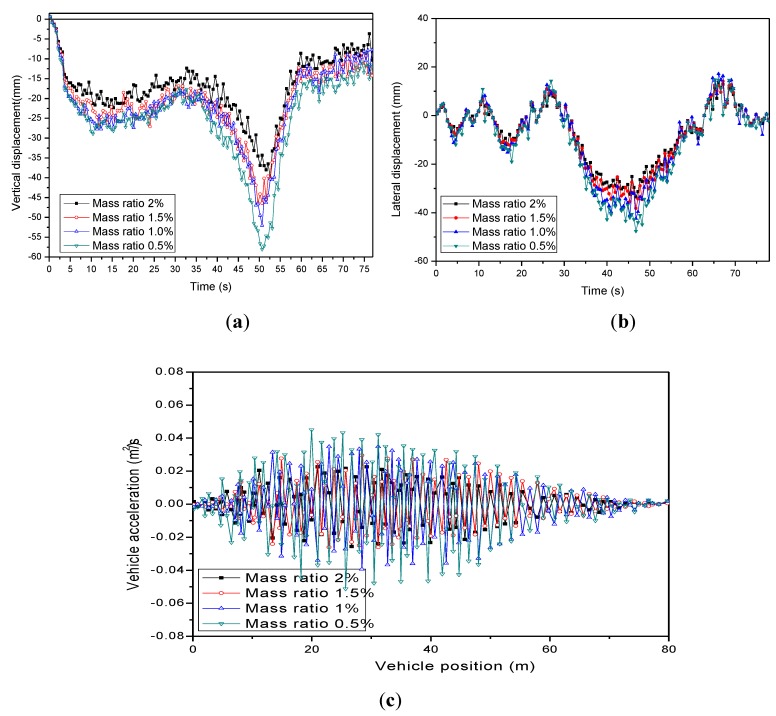
Parameters study of mass ratio on vehicle-bridge coupled vibration. (**a**) Vertical displacement. (**b**) Lateral displacement. (**c**) Vertical accelerations. (**d**) Lateral accelerations.

**Figure 14 sensors-19-01133-f014:**
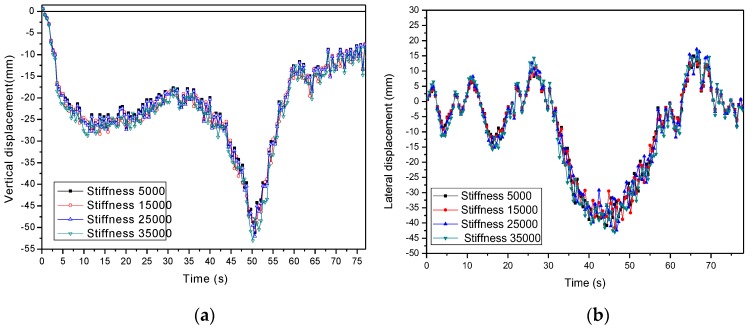
Parameters study of pounding stiffness on bridge displacements. (**a**) Vertical displacement. (**b**) Lateral displacement.

**Figure 15 sensors-19-01133-f015:**
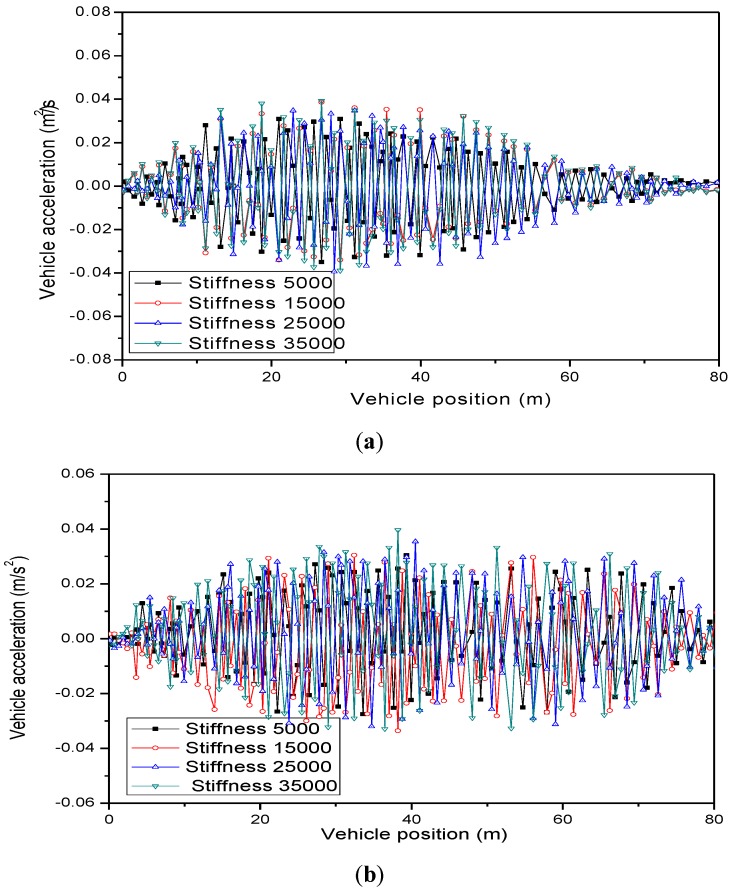
Parameters study of pounding stiffness on vehicle accelerations. (**a**) Vertical accelerations. (**b**) Lateral accelerations.

**Table 1 sensors-19-01133-t001:** Ride comfort standard specified in ISO2631-1.

$\mathrm{Vibration}\;\mathrm{Acceleration}\;\mathrm{Magnitudes}\;a_{w}$	Comfort or Discomfort
Less than 0.315 m/s^2^	Not uncomfortable
0.315 m/s^2^ to 0.63 m/s^2^	A little uncomfortable
0.5 m/s^2^ to 1 m/s^2^	Fairly uncomfortable
0.8 m/s^2^ to 1.6 m/s^2^	Uncomfortable
1.25 m/s^2^ to 2.5 m/s^2^	Very uncomfortable
Greater than 2 m/s^2^	Extremely uncomfortable

**Table 2 sensors-19-01133-t002:** Parameters of the 3-D vehicle.

Parameters	Unit
Mass of truck body *m_t_*	26,745 kg
Pitching moment of inertia of truck body *I_zt_*	162,650 kg.m^2^
Rolling moment of inertia of truck body *I_xt_*	67,656 kg.m^2^
Mass of truck front axle *m_a1_*	1513 kg
Rolling moment of inertia of front axle *I_xa1_*	2360 kg.m^2^
Mass of truck rear axle *m_a2_*	2674 kg
Rolling moment of inertia of rear axle *I_xa2_*	2360 kg.m^2^
Suspension spring stiffness of the first axle *K_sy_^1^, K_sy_^2^*	252,604 (N/m)
Suspension damper coefficient of the first axle *D_sy_^1^_,_ D_sy_^2^*	2490 (N.s/m)
Suspension spring stiffness of the second axle *K_sy_^3^, K_sy_^4^*	1,806,172 (N/m)
Suspension damper coefficient of the second axle *D_sy_^3^_,_ D_sy_^4^*	7982 (N.s/m)
Radial direction spring stiffness of the tire *k_ty_*	276,770 (N/m)
Radial direction spring damper coefficient of the tire *c_ty_*	1990 (N.s/m)
Length of the patch contact	345 mm
Width of the patch contact	240 mm
Distance between the front and rear axles *l_1_*	4.85 m
Distance between the front and the center of the truck *l_2_*	3.73 m
Distance between the rear axle and the center of the truck *l_3_*	1.12 m
Distance between the right and left axles *s_1_,s_2_*	2.40 m

**Table 3 sensors-19-01133-t003:** The parameters of the quarter vehicle model.

Parameters	Unit	Sedan Car	Light Truck
Sprung mass	kg	1611	4870
Stiffness of suspension system (*K_sx_^1^, K_sy_^1^, K_sz_^1^*)	N/m	434,920	500,000
Damping (*C_sx_^1^, C_sy_^1^, C_sz_^1^*)	N.s/m	5820	20,000

**Table 4 sensors-19-01133-t004:** Statistical property of traffic flow on bridge.

Occupancy	Average Speed (km/h)	Standard Deviation (km/h)
0.07	94.31	15.58
0.15	85.56	24.42
0.30	50.32	39.76

**Table 5 sensors-19-01133-t005:** Multiple tuned mass damper parameters.

Longitudinal Locations of PTMDs, *n*	(II-II)	(I-I; II-II; III-III)	(I-I; II-II; III-III)	(I-I; II-II; III-III)
Number of PTMDs, *n*	1	3 (One PTMD of each section)	6 (Two PTMDs of each section)	9 (Three PTMDs of each section)
Optimal frequency ratio	1.0	0.95;1.0;1.1	0.90;0.95;1.0; 1.05;1.1;1.15	0.85;0.88;0.90;0.95;1.0 1.05;1.10;1.15;1.20

**Table 6 sensors-19-01133-t006:** Maximum response of bridges with and without PTMD systems.

MTMD Condition	Dynamic Responses			
	Vertical Deflection (mm)	Reduction Ratio	Lateral Deflection(mm)	Reduction Ratio
Without PTMD	73.59		51.60	
Single PTMD	63.08	(14.28%)	46.71	(9.48%)
MPTMD(3)	52.02	(29.31%)	42.43	(17.77%)
MPTMD(6)	49.32	(32.98%)	39.42	(23.60%)
MPTMD(9)	46.52	(36.78%)	37.78	(26.78%)

**Table 7 sensors-19-01133-t007:** Ride comfort of vehicles with average roughness.

MTMD Condition	Dynamic Responses	
	*a*_su_ (m/s^2^)	Comfort or Discomfort
Without PTMD	1.04	Uncomfortable
Single PTMD	0.63	Fairly uncomfortable
MPTMD(3)	0.43	A little uncomfortable
MPTMD(6)	0.37	A little uncomfortable
MPTMD(9)	0.31	Not uncomfortable

**Table 8 sensors-19-01133-t008:** Maximum response of bridges with and without PTMD systems.

MTMD Condition	Dynamic Responses			
	Vertical Deflection (mm)	Reduction Ratio	Lateral Deflection(mm)	Reduction Ratio
Mass ratio 2.0%	40.13	(45.47%)	33.56	(37.98%)
Mass ratio 1.5%	46.41	(36.93%)	38.20	(25.71%)
Mass ratio 1.0%	52.02	(29.31%)	42.43	(21.59%)
Mass ratio 0.5%	58.07	(21.09%)	47.59	(12.05%)

**Table 9 sensors-19-01133-t009:** Ride comfort of vehicles with average roughness.

MTMD Condition	Dynamic Responses	
	*a*_su_ (m/s^2^)	Comfort or Discomfort
Mass ratio 2.0%	0.27	Not uncomfortable
Mass ratio 1.5%	0.32	A little uncomfortable
Mass ratio 1.0%	0.43	Fairly uncomfortable
Mass ratio 0.5%	0.56	Uncomfortable

**Table 10 sensors-19-01133-t010:** Maximum response of bridges with and without PTMD systems.

Stiffness Condition $\left({\mathrm{Nm}}^{-3/2}\right)$	Dynamic Responses			
	Vertical Deflection (mm)	Reduction Ratio	Lateral Deflection (mm)	Reduction Ratio
Stiffness 5000	50.62	(31.21%)	40.83	(24.54%)
Stiffness 15,000	51.12	(30.53%)	41.56	(23.19%)
Stiffness 25,000	52.02	(29.31%)	42.43	(21.59%)
Stiffness 35,000	53.03	(27.94%)	42.98	(20.57%)

**Table 11 sensors-19-01133-t011:** Ride comfort of vehicles with average roughness.

Stiffness Condition $\left({\mathrm{Nm}}^{-3/2}\right)$	Dynamic Responses	
	*a*_su_ (m/s^2^)	Comfort or Discomfort
Stiffness 5000	0.38	A little uncomfortable
Stiffness 15,000	0.41	A little uncomfortable
Stiffness 25,000	0.43	A little uncomfortable
Stiffness 35,000	0.46	A little uncomfortable

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
