# Peer review of "Vibration Suppression of Wind/Traffic/Bridge Coupled System Using Multiple Pounding Tuned Mass Dampers (MPTMD)"

_sensors, 2019, doi:10.3390/s19051133_

Round 1

Reviewer 1 Report

This is a very interesting work. The authors developed a comprehensive numerical simulation of the wind/traffic/bridge/Multiple PTMD coupled system with considering the effects of the traffic flow model, the MPTMD systems, and wind forces. Then, the parametric study of the different numbers and locations, mass ratio, and pounding stiffness of MPTMD were conducted respectively. Generally, this paper is well organized and written. It is therefore advised that the submission be accepted for publication subjected to minor revision.

General comments:

1. P2. The detailed vehicle parameters shown in Figures 1 and 2 should be given.

2. P9. How to determine the optimal frequency ratio of the MPTMD system?

3. P11, line 246~249. It is not easy to understand the statement, i.e., "the wind speed plays a significant role on the bridge displacements, especially for the lateral displacement". Actually, the vertical and lateral displacements are respectively increased by about 17 mm and 10 mm according to the results shown in Figure 11. Thus it seems that the vertical displacement is more sensitive to the wind speed than the lateral displacement.

4. In section 5.2, why the vertical displacement of the bridge is significantly increased due to wind forces in the lateral direction?

5. The lateral deflection of the bridge without PTMD is 54.11 mm (see Table 4), while the corresponding value given in line 249 (P11) is 51.60 mm. They are different, so please clarify.

6. From Tables 4, 6, and 8, it is found that generally the reduction ratios in the vertical direction are greater than that in the lateral direction. Please clarify the underlying causes.

Author Response

1. P2. The detailed vehicle parameters shown in Figures 1 and 2 should be given.

Answer 1: The vehicular parameters were added and shown in Table 2-3, and the references about the parameters were added in the revision.

2. P9. How to determine the optimal frequency ratio of the MPTMD system?

Answer 2: The optimal frequency ratio of the MPTMD system can be obtained as the previous study results and the experimental experience. For examples, in the studies (Jo et al., 2001; Li et al., 2005; Shi and Cai 2008), the mass ratio of the each PTMD is selected as 1%, the MPTMD system can own a wider frequency range, and each PTMD contains respective adjustable natural frequency.

3. P11, line 246~249. It is not easy to understand the statement, i.e., "the wind speed plays a significant role on the bridge displacements, especially for the lateral displacement". Actually, the vertical and lateral displacements are respectively increased by about 17 mm and 10 mm according to the results shown in Figure 11. Thus it seems that the vertical displacement is more sensitive to the wind speed than the lateral displacement.

Answer 3: In the revision, the “…especially for the lateral displacement” has been deleted.  

4. In section 5.2, why the vertical displacement of the bridge is significantly increased due to wind forces in the lateral direction?

Answer 4: The wind forces acting on the both vertical and lateral direction for the bridge-traffic coupled system. The vehicle vibration is significantly affected by the wind forces, and then this vibration can also affect the bridge vibration.

5. The lateral deflection of the bridge without PTMD is 54.11 mm (see Table 4), while the corresponding value given in line 249 (P11) is 51.60 mm. They are different, so please clarify.

Answer 5:  It has been modified in the revision, and 54.11 mm is not correct.

6. From Tables 4, 6, and 8, it is found that generally the reduction ratios in the vertical direction are greater than that in the lateral direction. Please clarify the underlying causes.

Answer 6: The reason is that the reduction efficiency of PTMD for the main structure in the vertical direction is greater than that in the lateral direction.

Reviewer 2 Report

In this manuscript the efficiency of multiple pounding mass dampers is analysed when these passive damping devices are used to control the wind and vehicle-induced bridge vibrations.  A sensitivity study of the number, the location, mass ratios and pounding stiffness of the multiple pounding mass dampers has been included.

The main contribution of the manuscript is the application of these well-known passive damping devices for the particular case of controlling the vibrations associated with the wind and traffic induced vibrations in bridges.

Although the study is really interesting and I am sure it will be interesting for the readers of the journal in its actual version presents several drawbacks that, according to this reviewer's opinion, need to be solved before to be considered for publication in this journal.

Major Aspects

There is a clear lack of information so that the manuscript can be followed and reproduced. It is supposed that both the traffic and the wind action have been defined under a stochastic approach. However they have not been defined adequately in the manuscript. Only some statistical parameters related to the traffic velocity have been included. The parameters that characterize each vehicle model should be included. Are these parameters deterministic or stochastic? Additionally, the wind action should be defined (deterministic or stochastic?).

In order to obtain the response of the structure under this stochastic excitation, some probabilistic or fuzzy logic method should be considered. The method should be described and the performance of the passive damping devices under these stochastic circumstances should be checked. Please clarify this aspect in the main manuscript.

Minor Aspects

Page 3 Lines 78-81, the reviewer considers that it is necessary to define the variable vectors,in relation with the velocity and acceleration.

Page 3 Line 82, the subtitle “2.3 Wind forces on traffic-flow bridge coupled system in model coordinates” is not adequate and perhaps even messy so the paragraph describes the equation of motion of the structure (the bridge). Please clarify this point.

Page 3 Line 90-92, the wind force is not an external force. Please clarify the definition of the different components of the force vector.

Page 5 Line 120, the different terms of Equation (5) should be defined.

The reviewer considers that section 2 and section 3 could be simplified in a unique section where the MPTMD-vehicule-bridge interaction model was described.

Author Response

Major Aspects

1. There is a clear lack of information so that the manuscript can be followed and reproduced. It is supposed that both the traffic and the wind action have been defined under a stochastic approach. However they have not been defined adequately in the manuscript. Only some statistical parameters related to the traffic velocity have been included. The parameters that characterize each vehicle model should be included. Are these parameters deterministic or stochastic? Additionally, the wind action should be defined (deterministic or stochastic?).

Answer 1: In the revision, some details were added and given some references.

In the present study, to simplify the vehicular model, only heavy trucks are modeled with 3-D vehicle models while light trucks and sedan cars are modeled using quarter vehicle models to save computational efforts. The 3-D vehicle model and the quarter vehicle model are shown in Fig.1 and Fig.2, respectively, and the parameters of the vehicle models are summarized in Tables 2 and 3. The mechanical and geometric properties of the test truck can be obtained from Yin et al. (2011) and are listed in Table 2. The parameters of the quarter vehicle model can be obtained from Chen and Cai (2004) and are also shown in Table 3.

For the purpose of traffic simulation, three different vehicle occupancy coefficients are considered (Chen and Wu 2011) using the Eqs. (1-3): smooth traffic (), median traffic (), and busy traffic flow (). It is reasonably found from Table 4 that the mean speed of the traffic flow decreases while the standard deviation of the vehicle speeds increases with the increase of the vehicle occupancy.

2. In order to obtain the response of the structure under this stochastic excitation, some probabilistic or fuzzy logic method should be considered. The method should be described and the performance of the passive damping devices under these stochastic circumstances should be checked. Please clarify this aspect in the main manuscript.

Answer 2: In the previous studies (Jo et al., 2001; Li et al., 2005; Shi and Cai 2008), the damping ratios of all the bridges are assumed to be 0.02. For the PTMD,  is the impact damping ratio correlated with the coefficient of restitution e, which is defined as the relation between the post impact (final) relative velocities. When e=0 stands for a perfectly plastic impact, e=1 stands for a fully elastic impact.

Minor Aspects

1. Page 3 Lines 78-81, the reviewer considers that it is necessary to define the variable vectors, in relation with the velocity and acceleration.

Answer 1: The variable vectors of velocity and acceleration are added in the revision.

2. Page 3 Line 82, the subtitle “2.3 Wind forces on traffic-flow bridge coupled system in model coordinates” is not adequate and perhaps even messy so the paragraph describes the equation of motion of the structure (the bridge). Please clarify this point.

Answer 2: the subtitle has been modified as the Equation of motion of the vehicle

3. Page 3 Line 90-92, the wind force is not an external force. Please clarify the definition of the different components of the force vector.

Answer 3: In the equation 2, is the vector of the wind forces acting on the bridge.

4. Page 5 Line 120, the different terms of Equation (5) should be defined.

Answer 4: The equation 5 has been defined in the revision. , , and  are the mass, damping, and stiffness matrices of the PTMD, respectively; , , is the displacement, acceleration, and velocity vector for all DOFs of the PTMD.

5. The reviewer considers that section 2 and section 3 could be simplified in a unique section where the MPTMD-vehicule-bridge interaction model was described.

Answer 5: in the revision, the section 2 and section 3 has been merged as a unique section.

Reviewer 3 Report

The manuscript deals with a vibration control system termed Multiple Pounding Tuned Mass Damper (MPTMD), proposed to suppress wind and traffic vibrations in bridge structures. The PTMD system combines the classical TMD with a delimiter covered with a viscoelastic material to restrict the stroke of the mass block and dissipate energy via impacts or pounding.

The authors presented the coupled equations of motion for describing the wind/traffic/bridge structure with the proposed MPTMD system. The Hertz contact element combined with a damper is used to simulate the nonlinear pounding force. Then, application to a high-pier bridge is presented with implementation of the proposed system, and the performance is assessed in terms of root-mean-square (RMS) value of the acceleration as per ISO2631-1(1997). An extensive parametric study was conducted to study the system performance under different flow occupancies and wind speeds, and to investigate the effect of numbers and locations of the MPTMD system, mass ratio and pounding stiffness. Based on this study, the authors have drawn some useful conclusions.

The paper is interesting and well written and certainly provides some original contributions to the problem under investigation. I would re-consider the paper for publication after the following points are addressed in a revised version:

-        Introduction, pag. 2 line 47:

In addition to the PTMD, there are a number of TMD variants that have been recently developed in the recent literature, which the authors should mention in the Introduction for a more complete overview of vibration control systems.

Most of them could be usefully employed to suppress vibrations in bridges and related applications, which is pertinent to the present study. Therefore, I recommend the authors integrate the bibliography with the following references:

1) tuned inerter damper and tuned mass damper inerter (TID and TMDI, respectively), which exploit the mass-amplification effects of the inerter, a mechanical two-terminal device whose generated force is proportional to the relative acceleration of its two terminals. This device has opened a new fruitful line of research in the field of more effective alternatives to large-mass ratio TMDs:

https://doi.org/10.1016/j.probengmech.2014.03.007

https://doi.org/10.1016/j.soildyn.2017.11.023

https://doi.org/10.1002/eqe.2390

http://dx.doi.org/10.1002/eqe.3011

2) combination of TMD with base isolation for improved performance against seismic loads:

https://doi.org/10.1016/0020-7683(94)00150-U

https://doi.org/10.1016/j.engstruct.2008.05.027

https://doi.org/10.1016/j.soildyn.2018.06.022

3) multi-TMD and distirbuted multi-TMD for wind induced response mitigation

https://doi.org/10.1002/stc.2275

http://dx.doi.org/10.1155/2014/198719

-        Introduction, pag. 2, line 50: robustness of TMD systems with large mass ratios was thoroughly investigated in other papers not mentioned here:

1)     Reggio, A., & Angelis, M. D. (2015). Optimal energybased seismic design of nonconventional Tuned Mass Damper (TMD) implemented via interstory isolation. Earthquake Engineering & Structural Dynamics, 44(10), 1623-1642.

2)     De Domenico, D., & Ricciardi, G. (2018). Optimal design and seismic performance of tuned mass damper inerter (TMDI) for structures with nonlinear base isolation systems. Earthquake Engineering & Structural Dynamics, 47(12), 2539-2560.

-        Pag. 3 line 81: please add more details on how the vector of wind forces acting on the vehicle has been evaluated in this study, by quoting some pertinent references/standards, if any;

-        Pag. 6 line 141: considering the nonlinearity effects induced by the pounding, which numerical algorithm did the author use to handle the nonlinearity of the equations of motion? Did they perform a sensitivity study by varying the time step to ensure that the response is accurate enough?

-        Pag. 7, Eq. (11): please check again this equations, as the symbols appear a bit distorted in the pdf;

-        Pag. 7, line 183. the authors should add some details about how they used the finite element (FE) model of the bridge shown in Fig. 8 for the present case study. Which data did they extract from the FE model?

-        Pag. 8, line 195: please motivate, with appropriate references/guidelines or standards, the choice of the three \rho values for simulating three different scenarios of traffic flow.

-        Pag. 9, line 218: please justify the value of \beta=25000 for the preliminary analysis;

-        Pag. 9, line 219: the authors should better explain how they identified the optimal frequency ratios shown in Table 3 for the different configurations of the PTMD analyzed in this study.

Author Response

Answer 1: most of the above suggested references were added in the revision.

2.       Pag. 3 line 81: please add more details on how the vector of wind forces acting on the vehicle has been evaluated in this study, by quoting some pertinent references/standards, if any;

Answer 2: The reference about the wind forces acting on the vehicle was added in the revision.

3.     Pag. 6 line 141: considering the nonlinearity effects induced by the pounding, which numerical algorithm did the author use to handle the nonlinearity of the equations of motion? Did they perform a sensitivity study by varying the time step to ensure that the response is accurate enough?

Answer 3:  The equations of traffic-bridge coupled system with MPTMD can be solved using the improved New-mark method, which can vary the time step to ensure the more accurate responses in the time domain.

4.       Pag. 7, Eq. (11): please check again this equations, as the symbols appear a bit distorted in the pdf;

Answer 4: the equation (11) has been modified in the revision. 

5.       Pag. 7, line 183. the authors should add some details about how they used the finite element (FE) model of the bridge shown in Fig. 8 for the present case study. Which data did they extract from the FE model?

Answer 5: The data about frequencies and modal of the FE model were used in analysis the vehicle-bridge coupled system.

6.        Pag. 8, line 195: please motivate, with appropriate references/guidelines or standards, the choice of the three \rho values for simulating three different scenarios of traffic flow.

Answer 6: Three different scenarios of traffic flow were obtained from the previous studies.

7.        Pag. 9, line 218: please justify the value of \beta=25000 for the preliminary analysis;

Answer 7: The parameter of beta with 2500 is referred from the references (Jo et al., 2001; Li et al., 2005; Shi and Cai 2008).

8.        Pag. 9, line 219: the authors should better explain how they identified the optimal frequency ratios shown in Table 3 for the different configurations of the PTMD analyzed in this study.

Answer 8: the date including the frequency ratios shown in Table 3 are all obtained from the previous studies and the experimental experiences. 

Round 2

Reviewer 3 Report

All the technical amendments suggested in the previous review have been meet by the authors satisfactorily. Therefore, the paper is now recommended for publication.